# Probabilistic Transformer for Time Series Analysis

**Binh Tang**
Department of Statistics and Data Science
Cornell University
Ithaca, NY 14850
bvt5@cornell.edu

**David S. Matteson**
Department of Statistics and Data Science
Cornell University
Ithaca, NY 14850
matteson@cornell.edu

## Abstract

Generative modeling of multivariate time series has remained challenging partly due to the complex, non-deterministic dynamics across long-distance time steps. In this paper, we propose deep probabilistic methods that combine state-space models (SSMs) with transformer architectures. In contrast to previously proposed SSMs, our approaches use attention mechanism to model non-Markovian dynamics in the latent space and avoid recurrent neural networks entirely. We also extend our models to include several layers of stochastic variables organized in a hierarchy for further expressiveness. Compared to transformer models, ours are probabilistic, non-autoregressive, and capable of generating diverse long-term forecasts with accounted uncertainty. Extensive experiments show that our models consistently outperform competitive baselines on various tasks and datasets, including time series forecasting and human motion prediction.

## 1 Introduction

Generative modeling of multivariate time series is a challenging problem with wide-ranging applications in demand forecasting [15, 76], autonomous driving [2, 16], robotics [29, 67], and health care [20, 21, 59]. Despite remarkable progress in recent years, models that predict high-dimensional future observations from a few past examples have remained intractable, partly due to the complex, non-deterministic temporal dynamics across long-distance time steps. Given a sequence of human poses, for example, such models must internally figure out the involved dynamics of various body components across space and time while maintaining the inherent uncertainty of multiple plausible futures, even though only one such future is observed.

Among proposed probabilistic approaches, state space models (SSMs) provide a principled framework for learning and drawing inference from sequential inputs [27, 66]. While autoregressive models feed its predictions back into the dynamics model without any compressed representation of data, SSMs model stochastic transitions between abstract states using latent variables, allowing for efficient state-to-state sampling without the need to render high-dimensional observations. Gaussian linear dynamical systems (LDSs), one of the best known SSMs [92], for example, postulate linear state transitions and enjoy exact inference via the celebrated Kalman filter algorithm.

While early extensions of LDSs focus on linearization [46] and unscented transform [88], recent work that marry state space models with deep neural networks offers much more flexibility to model complex dependencies across different time steps. Some approaches retain the Markovian dynamics of LDSs and only replace their linear observation models with feed-forward networks [23, 31, 47, 71], whereas others favor nonlinear state transitions and parametrize such dependencies via recurrent neural networks (RNNs) [22, 23, 30, 39, 51, 75]. Despite differences, both Markovian transitions and RNNs are often not capable of capturing long-range dependencies in highly structured sequential inputs [36, 100], limiting the capacity of the corresponding SSMs.

35th Conference on Neural Information Processing Systems (NeurIPS 2021).

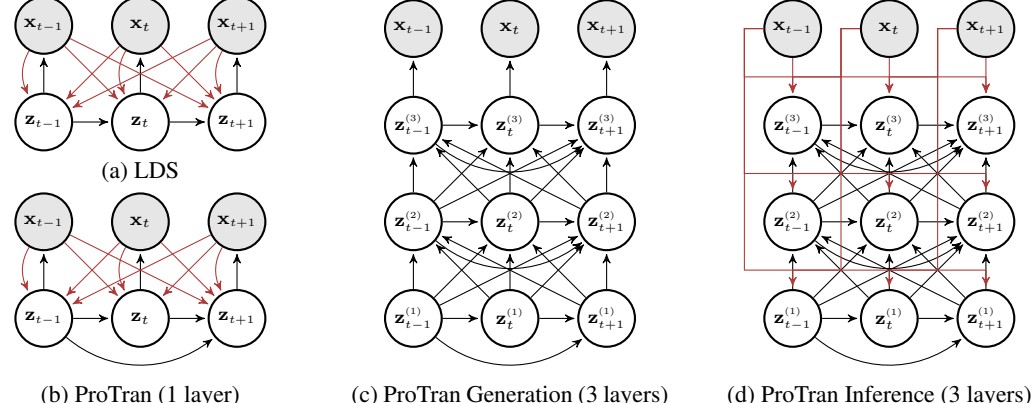

(a) LDS

(b) ProTran (1 layer)  (c) ProTran Generation (3 layers)  (d) ProTran Inference (3 layers)

Figure 1: Graphical model representations of linear dynamical systems (LDSs) in (a), and our proposed models (ProTran) in (b), (c), and (d). Black arrows denote the generative mechanism and red arrows the inference procedure. The separation of generation and inference in (c) and (d) is for readability. While traditional SSMs such as LDSs are limited to Markovian dynamics and linear dependencies, our models allow for non-Markovian and non-linear interactions between time steps via attention mechanism. A multi-layer extension of our models further increases expressiveness without compromising the tractable inference procedure.

In this work, we propose to combine the complementary strengths of SSMs and transformer architectures [85], a powerful mechanism for modeling long-term interactions that enjoys success across a variety of sequence modeling tasks [26, 48, 99]. In contrast to most SSMs, our models make extensive use of attention mechanism [5, 85] between latent variables to model non-Markovian dynamics (see Figure 1). Compared to transformer-based methods, our models are probabilistic, non-autoregressive in a similar fashion to LDSs, and capable of generating diverse long-term forecasts with uncertainty estimates.

Our main contributions are threefold. First, we propose novel SSMs based on transformer architectures for multivariate time series, which include generative models and inference procedures based on variational inference [49, 74]. Second, we extend our models to include several layers of stochastic latent variables organized in a hierarchy for further expressiveness. Third, we conduct extensive experiments on time series forecasting and human motion prediction and demonstrate that our Probabilistic Transformer (ProTran) performs remarkably well compared to various state-of-the-art baselines.

## 2 Preliminaries

### 2.1 Variational State Space Models

Let $\{\mathbf{x}_{1:T_i}^{(i)}\}_{i=1}^N$ consist of $N$ univariate time series where $\mathbf{x}_{1:T_i}^{(i)} = (x_1^{(i)}, x_2^{(i)}, \cdots x_{T_i}^{(i)})$ and $x_t^{(i)}$ denotes the vaue of the $i$-th time series at time $t$. We consider the multivariate form $\mathbf{x}_{1:T} = (\mathbf{x}_1, \mathbf{x}_2, \ldots, \mathbf{x}_T)$ where $\mathbf{x}_t = (x_t^{(1)}, \ldots x_t^{(N)}) \in \mathbb{R}^N$. Conditioning on observed values up to time $C$, we aim to produce distributional forecasts into the future $p(\mathbf{x}_{C+1:T} \mid \mathbf{x}_{1:C})$. For clarity, we refer to $\mathbf{x}_{1:C}$ and $\mathbf{x}_{C+1:T}$ as contexts and targets, respectively.

We are interested in probabilistic models parametrized by $\theta$ of the form

$$p_\theta(\mathbf{x}_{1:T} \mid \mathbf{x}_{1:C}) = \int p_\theta(\mathbf{x}_{1:T} \mid \mathbf{z}_{1:T}) p_\theta(\mathbf{z}_{1:T} \mid \mathbf{x}_{1:C}) d\mathbf{z}_{1:T} \tag{1}$$

where $\mathbf{z}_{1:T} = (\mathbf{z}_1, \mathbf{z}_2, \ldots, \mathbf{z}_T)$ denotes the corresponding sequence of latent variables, sometimes referred to as states. In other words, we assume a generative model that can be decomposed into a transition model $p_\theta(\mathbf{z}_{1:T} \mid \mathbf{x}_{1:C})$ between the latent variables conditioned on the contexts, and an emission model $p_\theta(\mathbf{x}_{1:T} \mid \mathbf{z}_{1:T})$ from the latent variables to observable outputs. In particular, we

further impose several assumptions on both models: [1]

$$p_\theta(\mathbf{z}_{1:T} \mid \mathbf{x}_{1:C}) = \prod_{t=1}^{T} p_\theta(\mathbf{z}_t \mid \mathbf{z}_{1:t-1}, \mathbf{x}_{1:C}), \qquad p_\theta(\mathbf{x}_{1:T} \mid \mathbf{z}_{1:T}) = \prod_{t=1}^{T} p_\theta(\mathbf{x}_t \mid \mathbf{z}_t). \qquad (2)$$

As demonstrated in Figure 1(b), the latent variable $\mathbf{z}_{t+1}$ depends not only on $\mathbf{z}_t$ but also on all of its preceding latent variables, including $\mathbf{z}_{t-1}$, in contrast to linear dynamical systems (LDSs). In addition, the transition and emission models allow for non-linearity via neural network parametrizations. These assumptions aim to maximize model capacity for real-world applications with complex emissions or temporal dependencies.

However, neither $\mathbf{x}_{1:t-1}$ nor $\mathbf{z}_{1:t-1}$ are included in the emission model $p(\mathbf{x}_t \mid \mathbf{z}_{1:T}, \mathbf{x}_{1:C})$. Such assumptions are important, as it has been argued previously that a leakage of information from the latent space in autoregressive models can hinder long-term predictions [23, 47]. While all ground truth observations are available during training, the entire sequence has to be generated sequentially at test time, making the dependencies on $\mathbf{x}_{1:t-1}$ prone to accumulated errors over multiple time steps. By letting the latent variable $\mathbf{z}_t$ capture all information needed to render $\mathbf{x}_t$, we also avoid the computational costs associated with repeatedly decoding and encoding $\mathbf{x}_t$ in multi-step predictions.

The inclusion of nonlinear state transitions and observation models necessarily requires approximate inference. We follow the stochastic variational inference framework [49, 74] and assume that the variational posterior parametrized by $\phi$ can be decomposed auto-regressively as $q_\phi(\mathbf{z}_{1:T} \mid \mathbf{x}_{1:T}) = \prod_t q_\phi(\mathbf{z}_t \mid \mathbf{z}_{1:t-1}, \mathbf{x}_{1:T})$, which leads to a lower bound on the log likelihood:

$$\log p_\theta(\mathbf{x}_{1:T} \mid \mathbf{x}_{1:C}) \geq \sum_{t=1}^{T} \left( \mathbb{E}_q \left[ \log p_\theta(\mathbf{x}_t \mid \mathbf{z}_t) \right] - \mathsf{KL}(q_\phi(\mathbf{z}_t \mid \mathbf{z}_{1:t-1}, \mathbf{x}_{1:T}) \parallel p_\theta(\mathbf{z}_t \mid \mathbf{z}_{1:t-1}, \mathbf{x}_{1:C})) \right),$$
$$(3)$$

where KL is the Kullback-Leibler divergence.

For computational stability, we assume homoscedasticity and choose Laplace distribution with scale parameter $\beta$ as a parametric form for $p_\theta(\mathbf{x}_t \mid \mathbf{z}_t)$, i.e. we optimize for $L_1$ reconstruction loss with a cross-validated factor $\beta$ for the KL term, following similar variational autoencoder (VAE) work [24, 41, 86]. Such an assumption does not necessarily limit the capacity of our models, as powerful stochastic transitions and flexible emission models can theoretically characterize arbitrary noise covariance [66]. Incorporating structured probabilistic outputs such as Gaussian copulas [75] or normalizing flows [23] can potentially further improve our model performance.

## 2.2 Transformer Architectures

Central to our models and other transformer-based approaches [48, 85] is the notion of attention [5], which allows the models to focus on important parts within a context. Multi-head attention, for example, maps a sequence of queries $\mathbf{Q} \in \mathbb{R}^{\ell_q \times d}$ of length $\ell_q$ to a sequence of outputs $\mathbf{O} = [\mathbf{O}_1, \ldots, \mathbf{O}_H] \in \mathbb{R}^{\ell_q \times d}$ of the same size by attending over given $\ell_k$ key-value pairs $\mathbf{K} \in \mathbb{R}^{\ell_k \times d}$, $\mathbf{V} \in \mathbb{R}^{\ell_k \times d}$:

$$\mathbf{O}_h = \mathsf{Attention}(\mathbf{Q}_h, \mathbf{K}_h, \mathbf{V}_h) = \mathsf{Softmax}\left(\frac{\mathbf{Q}_h \mathbf{K}_h^\mathsf{T}}{\sqrt{d}}\right) \mathbf{V}_h, \qquad (4)$$

where $\mathbf{Q}_h = \mathbf{Q}\mathbf{W}_h^Q$, $\mathbf{K}_h = \mathbf{K}\mathbf{W}_h^K$, $\mathbf{V}_h = \mathbf{V}\mathbf{W}_h^V$ are projected queries, keys, and values corresponding to head $h \in [1, H]$ with learning parameters $\mathbf{W}_h^Q, \mathbf{W}_h^K, \mathbf{W}_h^V$, respectively. In case $\mathbf{Q} = \mathbf{K} = \mathbf{V}$, we refer to such an attention mechanism as self-attention.

Given fully observed sequences of inputs, the mapping can be computed efficiently without any imposed sequential order often seen in recurrent neural networks [19, 42]. More importantly, the direct connections between long-distance time steps are baked into the mechanism as information from previous time steps is easily accessible without being compressed into a fixed representation, easing optimization and learning of long-term dependencies [5, 85].

---

[1]For notational simplicity, we assume $\mathbf{x}_0 = \mathbf{z}_{1:0} = \varnothing$ and $p(\mathbf{z} \mid \mathbf{x}_0) = p(\mathbf{z})$.

Without recurrence, Transformer [85] encodes information about each time step $t$ with pred efined sinusoidal positional embeddings $\mathsf{Position}(t) = [p_t(1), \ldots, p_t(d)] \in \mathbb{R}^d$ where the $i$-th embedding is given by $p_t(i) = \sin(t \cdot c^{i/d})$ for even $i$ and $p_t(i) = \cos(t \cdot c^{i/d})$ for odd $i$ and $c$ is some large constant. Empirical results show that such positional embeddings are also important to our models.

# 3 Probabilistic Transformer

In this section, we first present our single-layered model and subsequently its multi-layered extension for a hierarchy of stochastic latent variables. As alluded earlier, our model consists of a generative model and an inference model that share information and parameters extensively.

## 3.1 Single-Layered Probabilistic Transformer

**Generative Model.** Given some contexts $\mathbf{x}_{1:C}$, we first apply a linear projection and combine it with a positional embedding to obtain $\mathbf{h}_{1:C} \in \mathbb{R}^d$, i.e.

$$\mathbf{h}_t = \mathsf{LayerNorm}(\mathsf{MLP}(\mathbf{x}_t) + \mathsf{Position}(t)), \tag{5}$$

where LayerNorm and MLP denote layer normalizations [4] and multi-layer perceptrons, respectively. While a traditional transformer model often dedicates an entire encoder for the same purpose [55, 72], we find such a simple mapping works sufficiently well in conjunction with the context-attention module of the corresponding decoder.

As implied in Equation (2), our latent dynamics decomposes auto-regressively. At each time step, we parametri ze the distribution $p_\theta(\mathbf{z}_t \mid \mathbf{z}_{1:t-1}, \mathbf{x}_{1:C})$ by a Gaussian with parameters resulting from two sequential steps of attention: a self-attention over the previously inferred states $\mathbf{z}_{1:t-1}$ and another attention over the projected contexts $\mathbf{h}_{1:C}$. These two operations mirror those found in the decoder of Transformer [85], with the stochastic latent variables replacing its decoder inputs.

Unfortunately, using stochastic samples of $\mathbf{z}_t$ as attention queries is problematic, as purely stochastic transitions make it difficult for the model to reliably retain information across multiple time steps [17, 30, 39]. We therefore encapsulate the latent variables in hidden representations $\mathbf{w}_t$ that also has a deterministic component. Combined with the attention steps, such representations help model long-range temporal dependencies while accounting for the stochasticity of future observations.

Starting with a learnable, context-agnostic representation $\mathbf{w}_0$, we recursively update $\mathbf{w}_t$ using a stochastic sample from $p_\theta(\mathbf{z}_t \mid \mathbf{z}_{1:t-1}, \mathbf{x}_{1:C})$ and th e positional embedding for the current time step $t$. The generating process for the time step $t$ can be summarized by the following pseudocode:

$$\bar{\mathbf{w}}_t = \mathsf{LayerNorm}(\mathbf{w}_{t-1} + \mathsf{Attention}(\mathbf{w}_{t-1}, \mathbf{w}_{1:t-1}, \mathbf{w}_{1:t-1})) \tag{6}$$

$$\hat{\mathbf{w}}_t = \mathsf{LayerNorm}(\bar{\mathbf{w}}_t + \mathsf{Attention}(\bar{\mathbf{w}}_t, \mathbf{h}_{1:C}, \mathbf{h}_{1:C})) \tag{7}$$

$$\mathbf{z}_t = \mathsf{Sample}(\mathcal{N}(\mathbf{z}_t; \mathsf{MLP}(\hat{\mathbf{w}}_t), \mathsf{Softplus}(\mathsf{MLP}(\hat{\mathbf{w}}_t)))) \tag{8}$$

$$\mathbf{w}_t = \mathsf{LayerNorm}(\hat{\mathbf{w}}_t + \mathsf{MLP}(\mathbf{z}_t) + \mathsf{Position}(t)), \tag{9}$$

where Sample and Softplus are the Gaussian sampling and approximating rectifier operators.

Each stochastic sample of $\mathbf{w}_{1:T}$ is then mapped to a sequence of $\mathbf{x}_{1:T}$ via a multi-layer perceptron. We emphasize that our generation procedure in the latent space is more efficient than others in the observation space, which requires encoding and decoding high-dimensional inputs repeatedly.

**Inference Model.** We parametrize the approximate posterior $q_\phi(\mathbf{z}_t \mid \mathbf{z}_{1:t-1}, \mathbf{x}_{1:T})$ at time step $t$ in a simi lar fashion to the prior $p_\theta(\mathbf{z}_t \mid \mathbf{z}_{1:t-1}, \mathbf{x}_{1:C})$. Indeed, these parametrizations share most parameters and are done simultaneously in the same recursive loop, following the exact same steps in Equation (6) and Equation (7) (see Figure 1). We note that similar sharing techniques between the generative and inference processes have emerged as a common theme among recent successful VAE models [17, 62, 83].

While the prior only has access to the conditioning observations $\mathbf{x}_{1:C}$, the approximate posterior should take into account all observations during training, including the targets $x_{C+1:T}$. Due to the inherent unidirectional aspect of RNNs, previous work that uses RNNs to parametrize the approximate posterior often disregards such a property [22, 30, 51] and often resorts to a filtering routine

$p(\mathbf{z}_t \,|\, \mathbf{z}_{1:t-1}, \mathbf{x}_{1:t})$. In contrast, our inference procedure resembles more of the smoothing process of LDSs, factoring in both past and future observations via another application of self-attention:

$$\mathbf{k}_t = \mathsf{Attention}(\mathbf{h}_{1:T}, \mathbf{h}_{1:T}, \mathbf{h}_{1:T})) \tag{10}$$

$$\mathbf{z}_t = \mathsf{Sample}(\mathcal{N}(\mathbf{z}_t; \mathsf{MLP}([\hat{\mathbf{w}}_t, \mathbf{k}_t]), \; \mathsf{Softplus}(\mathsf{MLP}([\hat{\mathbf{w}}_t, \mathbf{k}_t])))). \tag{11}$$

Here, we replace Equation (8) in the generative model with Equation (11), where the hidden representation $\mathbf{k}_t$ summarizing all information relevant to the current tim estep $t$ has been concatenate to the latent-and-context-aware representation $\hat{\mathbf{w}}_t$ preceding the Gaussian parametrization.

The generative model and the inference model are trained end-to-end with a single stochastic variational inference objective stated in Equation (3). Such a variational bound includes the reconstruction loss for $\mathbf{x}_{1:C}$ and the KL term for $\mathbf{z}_{1:C}$. Alternatively, we can exclude these terms from the objective, which is equivalent to starting the inference process from $t = C + 1$ instead of $t = 1$.

Our models incur a time complexity of $\mathcal{O}(T^2 d)$ and a memory cost of $\mathcal{O}(T^2 d)$, where $T$ is the total sequence length and $d$ is the dimensionality of the latent space. The recursive latent dynamics also does not allow use the take full advantange of parallelizable attentions. However, we find that our models are still efficient in practice, especially for reasonably small values of $T$.

## 3.2   Multi-Layered Extension for Probabilistic Transformer

Inspired by recent work on hierarchical VAEs for non-sequential inputs [17, 80, 83, 101], we extend our proposed model to include several layers of latent variables, aiming to further increase its flexibility for modelling sequential data.

We represent each time step $t$ with a Ma rkov chain of $L$ latent variables $\mathbf{z}_t^{(1:L)} = (\mathbf{z}_t^{(1)}, \ldots, \mathbf{z}_t^{(L)})$ for simplicity (see Figure 1). The generative and inference model also decompose auto-regressively across different time steps and may exhibit non-Markovian dynamics:

$$p_\theta\left(\mathbf{x}_{1:T}, \mathbf{z}_{1:T}^{(1:L)} \,|\, \mathbf{x}_{1:C}\right) = \left(\prod_{t=1}^{T} p_\theta\left(\mathbf{x}_t \,|\, \mathbf{z}_t^{(L)}\right)\right) \left(\prod_{\ell=1}^{L} \prod_{t=1}^{T} p_\theta\left(\mathbf{z}_t^{(\ell)} \,|\, \mathbf{z}_{1:t-1}^{(\ell)}, \mathbf{z}_{1:T}^{(\ell-1)}, \mathbf{x}_{1:C}\right)\right) \tag{12}$$

$$q_\phi\left(\mathbf{z}_{1:T}^{(1:L)} \,|\, \mathbf{x}_{1:T}\right) = \prod_{\ell=1}^{L} \prod_{t=1}^{T} q_\phi\left(\mathbf{z}_t^{(\ell)} \,|\, \mathbf{z}_{1:t-1}^{(\ell)}, \mathbf{z}_{1:T}^{(\ell-1)}, \mathbf{x}_{1:T}\right). \tag{13}$$

Intuitively, we generate samples $\mathbf{x}_{1:T}$ conditioning on $\mathbf{x}_{1:C}$ by following the latent dynamics from the bottom up and using the generative process described earlier within each layer. Analogously, inference proceeds in the same order, resulting in a variational bound similar to Equation (3):

$$\log p_\theta(\mathbf{x}_{1:T} \,|\, \mathbf{x}_{1:C}) \geq \sum_{t=1}^{T} \mathbb{E}_q\left[\log p_\theta(\mathbf{x}_t^{(L)} \,|\, \mathbf{z}_t)\right] \tag{14}$$

$$- \sum_{\ell=1}^{L} \mathsf{KL}(q_\phi(\mathbf{z}_t^{(\ell)} \,|\, \mathbf{z}_{1:t-1}^{(\ell)}, \mathbf{z}_{1:T}^{(\ell)}, \mathbf{x}_{1:T}) \,\|\, p_\theta(\mathbf{z}_t^{(\ell)} \,|\, \mathbf{z}_{1:t-1}^{(\ell)}, \mathbf{z}_{1:T}^{(\ell)}, \mathbf{x}_{1:C})). \tag{15}$$

As before, we parametrize the prior $p_\theta(\mathbf{z}_t^{(\ell)} \,|\, \mathbf{z}_{1:t-1}^{(\ell)}, \mathbf{z}_{1:T}^{(\ell)}, \mathbf{x}_{1:C})$ using self-attention over the inferred latent variables from previous time steps $\mathbf{w}_{t-1}^{(\ell)}$ on the same layer and another attention over contexts $\mathbf{h}_{1:C}$. In this case, however, we include an additional self-attention over all latent variables from the layer immediately below it (see Equation (16)):

$$\tilde{\mathbf{w}}_t^{(\ell)} = \mathsf{LayerNorm}(\mathbf{w}_{t-1}^{(\ell)} + \mathsf{Attention}(\mathbf{w}_{t-1}^{(\ell)}, \mathbf{w}_{1:T}^{(\ell-1)}, \mathbf{w}_{1:T}^{(\ell-1)})) \tag{16}$$

$$\bar{\mathbf{w}}_t^{(\ell)} = \mathsf{LayerNorm}(\tilde{\mathbf{w}}_t^{(\ell)} + \mathsf{Attention}(\tilde{\mathbf{w}}_t^{(\ell)}, \mathbf{w}_{1:t-1}^{(\ell)}, \mathbf{w}_{1:t-1}^{(\ell)})) \tag{17}$$

$$\hat{\mathbf{w}}_t^{(\ell)} = \mathsf{LayerNorm}(\bar{\mathbf{w}}_t^{(\ell)} + \mathsf{Attention}(\bar{\mathbf{w}}_t^{(\ell)}, \mathbf{h}_{1:C}, \mathbf{h}_{1:C})) \tag{18}$$

$$\mathbf{z}_t^{(\ell)} = \mathsf{Sample}(\mathcal{N}(\mathbf{z}_t^{(\ell)}; \mathsf{MLP}(\hat{\mathbf{w}}_t^{(\ell)}), \; \mathsf{Softplus}(\mathsf{MLP}(\hat{\mathbf{w}}_t^{(\ell)})))) \tag{19}$$

$$\mathbf{w}_t^{(\ell)} = \mathsf{LayerNorm}(\hat{\mathbf{w}}_t^{(\ell)} + \mathsf{MLP}(\mathbf{z}_t^{(\ell)}) + \mathsf{Position}(t)), \tag{20}$$

Stacking multiple layers of latent variables increases model expressiveness, but it also result in a linear increase in running time and the number of parameters. The time complexity for the $L$-layers transformer is $\mathcal{O}(LT^2d)$, while the space complexity remains $\mathcal{D}(T^2d)$ due to the Markovian structure of the chain $\mathbf{z}_t^{(1:L)}$ at each time step $t$. In our experiments, we restrict the number of layers of our hierachical models to two or three.

## 4  Related Work

**Deep State Space Models.**   Deep neural networks have been extensively combined with state space models, resulting in flexible, yet principledly motivated latent variable approaches. While some work keep the linear state transition intact to leverage the efficient Kalman filer algorithms [23, 31, 47, 71], more expressive, nonlinear latent dynamics parametrized by neural networks have been proposed [51, 52]. All such models are limited to the Markovian dynamics of LDSs, which hinders learning of long-range dependencies. The limitation is often alleviated by combining the stochastic transitions with a deterministic RNN that enables access to all past states [3, 8, 22, 30, 39, 77]. Our models are similarly non-Markovian, but the dependencies on the past states are done via attention, which allows for easy connections between long-distance time steps. In addition, while most existing deep SSMs represent each time step with a single latent variable, our models include several layers of hierarchical latent variables with tractable inference mechanism.

**Attentive Recurrent Networks.**   Attention mechanism has also been widely adopted in recent time series work using sequence-to-sequence models [1, 28] or transformer architectures [14, 55, 57, 72, 81, 94]. While our models are equipped with latent variables, these transformer approaches [55, 72] lack inference mechanism and are susceptible to feeding back observation noise into the dynamics model at test time. Our work, however, can be considered as an extension of the attentive state space model proposed in [1], with discrete latent states replaced by their continuous analogs. Recent developments in natural language processing [58, 60, 90] also combine transformer and VAE; however, these approaches often use a time-agnostic latent variable, in contrast to our SSM formulation.

**Time Series Forecasting.**   Traditional univariate time series models, such as Box-Jenkins methods [12] and exponential smoothing [43], often assume independence between any collection of time series [76]. While multivariate extensions of the classical approaches, including vector autoregression [82] and multivariate GARCH [7], do not require such a strong assumption, they come with many others such as stationarity and homocesdasticity, demand manual selection of covariates and models, and do not scale well to even a moderate number of time series [40, 69].

Deep learning methods for time series forecasting have recently emerged as an expressive, scalable framework for industrial applications [10, 68, 79, 91]. While early work focus on point forecasts [53, 70, 96], recent approaches often employ recurrent neural networks with probabilistic forecasts parametrized directly [76], using quantile functions [33], Gaussian copulas [75], normalizing flows [23], or diffusion models [73]. In contrast, our models are entirely devoid of such recurrent architectures and rely on latent variables to output distributional forecasts.

**Human Motion Prediction.**   Despite being almost identical in formulation, human motion prediction has often been studied independently from time series forecasting. While some work deterministically generate future motions or video frames [13, 32, 34, 56], stochastic prediction has also been proposed with deep neural networks often outperforming traditional methods such as hidden Markov models [93] or Gaussian processes [89] on complex motion datasets [13, 32, 45, 54, 63]. In contrast to earlier work [95, 97] that employ a global latent variable across different time steps via conditional VAE [49], we leverage the principled framework of state space models for learning and inference of hierarchical, time-dependent latent variables.

## 5  Experiments

We present our experiment results on two tasks, namely, time series forecasting and human motion prediction. These tasks are often studied independently, despite being almost identical as conditional prediction problems.

Table 1: Test set CRPS$_{\mathsf{sum}}$ of time series forecasting models (lower is better). The means and standard deviations are computed over five runs using different random seeds.

| DATASET | SOLAR | ELECTRICITY | TRAFFIC | TAXI | WIKIPEDIA |
|---|---|---|---|---|---|
| VES [43] | $0.900 \pm 0.003$ | $0.880 \pm 0.004$ | $0.350 \pm 0.002$ | - | - |
| VAR [61] | $0.830 \pm 0.006$ | $0.039 \pm 0.001$ | $0.290 \pm 0.001$ | - | - |
| VAR-Lasso [61] | $0.510 \pm 0.006$ | $0.025 \pm 0.000$ | $0.150 \pm 0.002$ | - | $3.100 \pm 0.004$ |
| GARCH [84] | $0.880 \pm 0.002$ | $0.190 \pm 0.001$ | $0.370 \pm 0.001$ | - | - |
| DeepAR [76] | $0.336 \pm 0.014$ | $0.023 \pm 0.001$ | $0.055 \pm 0.003$ | - | $0.127 \pm 0.042$ |
| LSTM-Copula [75] | $0.319 \pm 0.011$ | $0.064 \pm 0.008$ | $0.103 \pm 0.006$ | $0.326 \pm 0.007$ | $0.241 \pm 0.003$ |
| GP-Copula [75] | $0.337 \pm 0.024$ | $0.024 \pm 0.002$ | $0.078 \pm 0.002$ | $0.208 \pm 0.183$ | $0.086 \pm 0.004$ |
| KVAE [51] | $0.340 \pm 0.025$ | $0.051 \pm 0.019$ | $0.100 \pm 0.005$ | - | $0.095 \pm 0.012$ |
| NKF [23] | $0.320 \pm 0.020$ | $\mathbf{0.016 \pm 0.001}$ | $0.100 \pm 0.002$ | - | $0.071 \pm 0.002$ |
| Transformer-MAF [72] | $0.301 \pm 0.014$ | $0.021 \pm 0.000$ | $0.056 \pm 0.001$ | $0.179 \pm 0.002$ | $0.063 \pm 0.003$ |
| TimeGrad [73] | $0.287 \pm 0.020$ | $0.021 \pm 0.001$ | $0.044 \pm 0.006$ | $0.114 \pm 0.020$ | $\mathbf{0.049 \pm 0.002}$ |
| **ProTran (Ours)** | $\mathbf{0.194 \pm 0.030}$ | $\mathbf{0.016 \pm 0.001}$ | $\mathbf{0.028 \pm 0.001}$ | $\mathbf{0.084 \pm 0.003}$ | $\mathbf{0.047 \pm 0.004}$ |

## 5.1 Time-series Forecasting

**Datasets & Covariates.** Following the experiment setup in [72, 73, 75], we evaluate our models and multiple competitive baselines on five popular public datasets: SOLAR, ELECTRICITY, TRAFFIC, TAXI, and WIKIPEDIA. The data is recorded with hourly or daily frequency and shows seasonal patterns of different frequencies (see Appendix A for more dataset details). As in [72, 73], the covariates include lagged inputs, fixed time embeddings (e.g. day of week, hour of day), and learnable time-series embeddings. The inputs are scaled using the conditioning examples before being fed into the model, and the predictions are rescaled appropriately afterward.

**Metrics.** Following [23, 72, 75], we evaluate our model and all baselines using *continuous ranked probability score* (CRPS) [65] summed across time series, denoted by CRPS$_{\mathsf{sum}}$. Given a univariate distribution function $F$ and an observation $x$, CRPS is defined as

$$\mathsf{CRPS}(F, x) = \int_{\mathbb{R}} (F(z) - 1_{\{x \le z\}})^2 dz,$$

where $1_{\{x \le z\}}$ is the indicator function. As argued in de Bézenac et al. [23], CRPS$_{\mathsf{sum}}$ is a proper scoring rule [35] and can be computed without analytical forecast distributions. We compute the metrics in a rolling fashion and use 100 samples for the distributional forecasts, similar to the aforementioned work.

**Baselines.** We benchmark our models against various baselines, including (1) VES [43], an innovation state space model; (2) VAR-Lasso and VAR [61], two multivariate linear autoregressive models with and without Lasso regularization; (3) GARCH [84], a multivariate conditional heteroskedastic model; (4) DeepAR [76], an autoregressive recurrent neural network; LSTM-Copula and GP-Copula [75], two RNN-based models that use Gaussian copula to model nonlinearity; (5) KVAE [51], a variational approach based on linear dynamics; (6) NKF [23], a normalizing-flow model coupled with Kalman filters; (7) Transformer [72], a transformer-based model based on masked autoregressive flow; and (8) TimeGrad [73], a recent autoregressive approach that uses a diffusion model.

**Implementations.** We use 8-head attentions and 2-layers MLPs to parametrize the generative and inference models. The stochastic latent variables $\mathbf{z}_t$ are 16-dimensional while the hidden representations $\mathbf{w}_t$ are in $\mathbb{R}^{128}$. Our probabilistic transformers for SOLAR and ELECTRICITY have one stochastic layer while those for the other datasets of higher dimensional observations employ two layers. We re-

Table 2: Ablation study on TRAFFIC.

| | | | | |
|---|---|---|---|---|
| Two Layers | ✓ | ✗ | ✗ | ✗ |
| One Layer | ✗ | ✓ | ✓ | ✓ |
| Context Attention | ✓ | ✓ | ✗ | ✓ |
| Deterministic | ✗ | ✗ | ✗ | ✓ |
| CRPS$_{\mathsf{sum}}$ | **0.028** | 0.031 | 0.033 | 0.041 |

port the numbers of parameters of our models in Table 4 in Appendix C, which are all comparable to those of the state-of-the-art approaches. See Appendix D for more details about hyper-parameters and training processes.

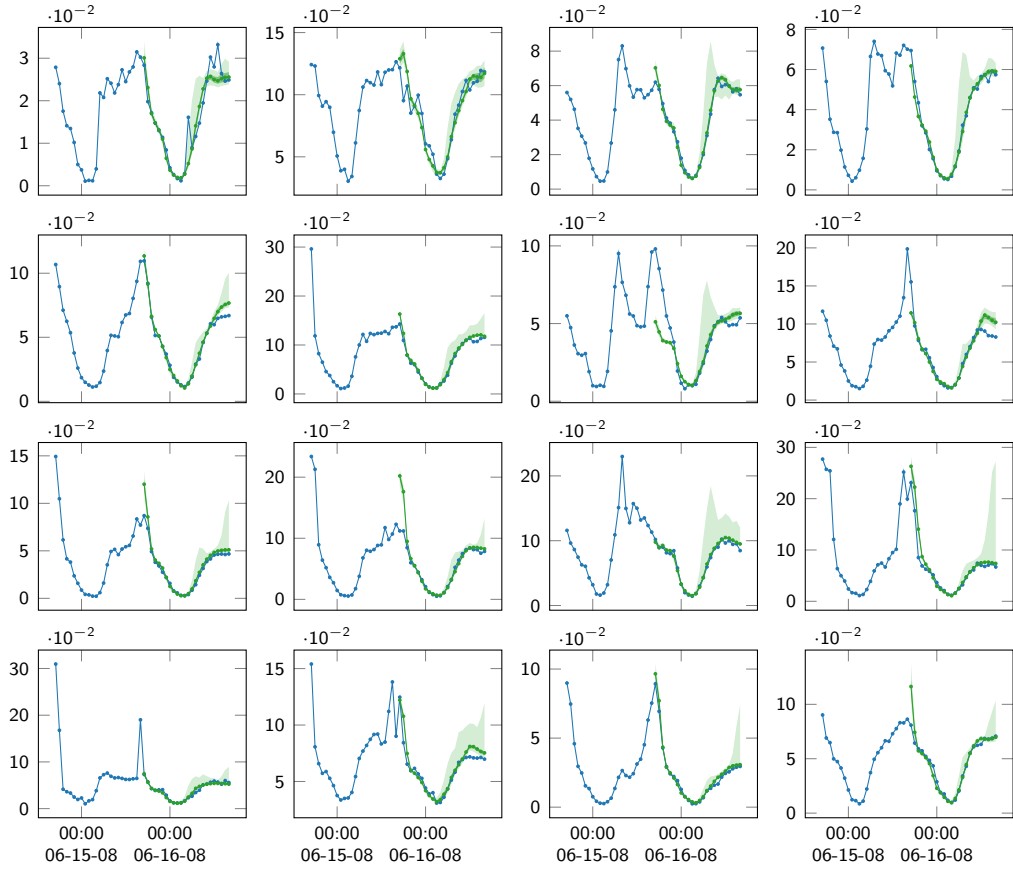

Figure 2: Prediction intervals and test set ground-truth from ProTran (our model) for the TRAFFIC dataset of the first 16 of 963 time series.

**Accuracy Comparison.** Table 1 shows that our models perform competitively across all five high-dimensional time series datasets, achieving CRPS$_{sum}$ comparable to the best methods on ELECTRIC-ITY and WIKIPEDIA while outperforming all baselines, including a non-SSM transformer-based approach [72], by significant margins on SOLAR, TRAFFIC and TAXI. Further analyses with other metrics, including CRPS and MSE, in Appendix B also help confirm our findings.

**Qualitative Results.** Figure 2 shows that the distribution forecasts generated by our model follow closely the ground truths, which is consistent with our accuracy results. In addition, the model appears to capture the uncertainty of future forecasts to some extent; observations of large magnitudes and far into the future seem to correctly have higher variance estimates.

**Ablation Study.** We include a small scale ablation study on the TRAFFIC dataset to investigate which components of our models are essential. Table 2 suggests that removing the stochasticity from $\mathbf{w}_t$ has most impacts on model performance, implying that incorporating latent variables into a transformer is indeed useful. Other aspects such as context attention or multiple layers of stochastic variables do not show dramatic effects in this study; however, they do contribute performance gains.

## 5.2 Human Motion Prediction

**Datasets.** Following the experiment setup in [97], we evaluation our models on two public motion capture datasets: Human3.6M[44] and HumanEva-I [78]. While Human3.6 is a large-scale dataset with 3.6 million video frames recorded at 50Hz, HumanEva-I is smaller with only 3 subjects and recorded at 60Hz. We follow the preprocessing steps of previous work [64, 97] and obtain a 17-joint skeleton for Human3.6 and a 15-joint skeleton for HumanEva-I. As in [97], we predict future motion for 2 seconds conditioning on observed motion of 0.5 seconds and 1 second conditioning on 0.25 seconds for Human3.6 and HumanEva-I, respectively.

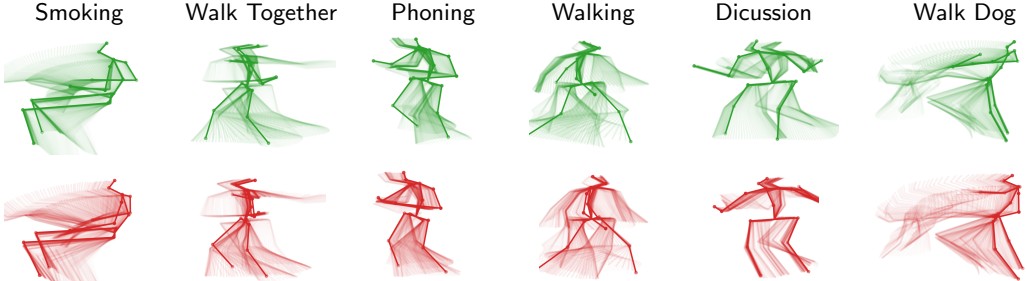

Figure 3: Ground-truth pose sequences (first row) and corresponding predictions by ProTran (second row). Solid colors indicate later time-steps and faded ones are older. The body-part movements in the predicted and ground-truth poses resemble similar patterns, while certain variations are retained.

Table 3: Human motion prediction results.

| DATASET | HUMAN3.6M | | HUMANEVA-I | |
|---------|-----------|---------|------------|---------|
| Method | ADE ↓ | FDE ↓ | ADE ↓ | FDE ↓ |
| ERD [32] | 0.722 | 0.969 | 0.382 | 0.461 |
| acLSTM [56] | 0.789 | 1.126 | 0.429 | 0.541 |
| MT-VAE [95] | 0.457 | 0.595 | 0.345 | 0.403 |
| Pose-Knows [87] | 0.461 | 0.560 | 0.269 | 0.296 |
| HP-GAN [6] | 0.858 | 0.867 | 0.772 | 0.749 |
| Best-Many [11] | 0.448 | 0.533 | 0.271 | 0.279 |
| GMVAE [25] | 0.461 | 0.555 | 0.305 | 0.345 |
| DeliGAN [38] | 0.483 | 0.534 | 0.306 | 0.322 |
| DSP [98] | 0.493 | 0.592 | 0.273 | 0.290 |
| DLow [97] | 0.425 | 0.518 | 0.251 | 0.268 |
| **ProTran (Ours)** | **0.381** | **0.491** | **0.258** | **0.255** |

**Metrics.** Following previous work on trajectory forecasting [2, 37], we adopt two popular metrics, namely, average displacement error (ADE) and final displacement error (FDE). ADE measures the average $L_2$ distance over all time steps between the ground truth motion and the closest sample, while FDE only consider such distance for the final pose.

**Baselines.** We compare our models against 9 models, including ERD [32] and acLSTM [56], two deterministic RNN-based approaches; MT-VAE [95] and Pose-Knows [87], two conditional VAE models; HP-GAN [6], a conditional GAN; Best-Many [11], GMVAE [25], DeliGAN [38]. and DSP [98], four approaches optimizing for diversity objectives. The results for these baselines are reported as in [97].

**Implementations.** Similar to the previous experiments, we use 8-head attentions and 2-layers MLPs. Since Human3.6M is significantly more complex and multi-modal than the time series forecasting datasets, we make use of 3 stochastic layers, as opposed to 2 layers for HumanEva-I. For Human3.6M, the context and target observations are significantly longer and set up for long-term predictions, so we only infer latent variables for target observations. Appendix C also contains further details about our models and their number of parameters.

**Quantitative Results.** Table 3 shows that our models convincingly outperform all baselines based on both metrics ADE and FDE, with the gains significantly higher for the larger dataset Human3.6M. We emphasize that our favorable performance is evaluated using random samples, while the closest competitor, DLow [97], relies on a separate model for selecting samples to promote diversity, which can potentially be combined with our probabilistic transformer for further improvements.

**Qualitative Results.** We show in Figure 3 human pose predictions made by our model that are most similar to the corresponding ground truths among a collection of such stochastic predictions. The similarities between the body-part movements in both sequences suggest that our model has been able to capture the temporal dynamics quite well.

# 6 Conclusion & Discussion

In this work, we have introduced generative models for multivariate time series that combines strengths of state space models and transformer architectures. In contrast to previous work, our models do not rely on recurrent neural networks but make extensive use of attention mechanism. We also extend our models to include hierarchical latent variables, inspired by recent developments of VAEs for non-sequential data [17, 83]. Empirical experiments show that our models perform remarkably well on time series forecasting and human motion prediction.

Our models do not come without limitations, however. As in other transformer-based approaches, the reliance on attention incurs a quadratic time and memory complexity. While we do not find it problematic in our experiments, the limitation necessarily hinders applications of our models in tasks characterized by long-term dependencies such as language modelling or music generation [36]. Fortunately, recent work on sparse transformer [9, 18, 50, 55] can potentially address the issue, and we leave such an investigation for future work.

Probabilistic time series forecasting is a fundamental research problem with wide-ranging applications in society. Although we have not explored healthcare applications of our work, previously proposed methods with similar formulations have demonstrated potentials of forecasting techniques [1, 81] in diagnoses or disease control.

# 7 Acknowledgment

Financial support is gratefully acknowledged from a Xerox PARC Faculty Research Award, National Science Foundation Awards 1455172, 1934985, 1940124, and 1940276, USAID, and Cornell University Atkinson Center for a Sustainable Future.

,

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
