# A Datasets

Following [5, 7], we run experiments on five popular datasets for time series forecasting, including SOLAR [3], ELECTRICITY [1], TRAFFIC [2], TAXI [3], and WIKIPEDIA [4]. Table 1 includes more details about the datasets.

Table 1: Dimension, domain, frequency, total training timesteps and prediction length properties of the training datasets used in the experiments.

| DATASET | DIMENSION | DOMAIN | FREQUENCY | TIMESTEPS | PREDICTION |
|---------|-----------|--------|-----------|-----------|------------|
| SOLAR | 137 | $\mathbb{R}^+$ | Hour | 7,009 | 24 |
| ELECTRICITY | 370 | $\mathbb{R}^+$ | Hour | 5,790 | 24 |
| TRAFFIC | 963 | $(0, 1)$ | Hour | 10,413 | 24 |
| TAXI | 1,214 | $\mathbb{N}$ | 30-Min | 1,488 | 24 |
| WIKIPEDIA | 2,000 | $\mathbb{N}$ | Day | 792 | 30 |

For human motion prediction, we run experiments on two datasets, namely, Human3.6M[2] and HumanEva-I [8], following [10]. As described in Section 5, Human3.6 is a large-scale dataset with 11 subjects performing 15 actions, totaling 3.6 million video frames recorded at 50Hz. To be consistent with previous work [4, 10], we adopt a 17-joint skeleton and train on 5 subjects (S1, S5, S6, S7, S8) and test on two subjects (S9, S11). For HumanEva-I, we adopt a 15-joint skeleton and use the same training and test split provided in the dataset. As in [10], we predict future motion for 2 seconds conditioning on observed motion of 0.5 seconds and 1 second conditioning on 0.25 seconds for Human3.6 and HumanEva-I, respectively.

Based on the descriptions of the datasets from previous work, we assume that they were obtained and curated appropriately with consent from pertaining people and that they contain no personally identifiable information or offensive content.

# B Additional Experiment Results

## B.1 Time Series Forecasting

In addition to $\text{CRPS}_{\text{sum}}$ reported in Section 5, we also include experiment results for time series forecasting using two other metrics, namely normalized root mean squared error ($\text{NRMSE}_{\text{sum}}$) and normalized deviation ($\text{ND}_{\text{sum}}$), in Table 2. As in [1], we define $\text{NRMSE}_{\text{sum}}$ as the root mean squared error normalized by the absolute values of targets summed across all time series. $\text{ND}_{\text{sum}}$, is defined as the mean absolute error between predicted values and targets summed across all time series.

Consistent with the results in Section 5, our models perform significantly better than Transformer-MAF [5] and TimeGrad [6], two competitive baselines proposed recently.

Table 2: Test set $\text{NMSE}_{\text{sum}}$ and $\text{CRPS}$ of time series forecasting models (lower is better). The means and standard deviations are computed over five runs using different random seeds.

| DATASET | SOLAR | | ELECTRICITY | | TRAFFIC | |
|---------|-------|---|-------------|---|---------|---|
| Method | $\text{NRMSE}_{\text{sum}}$ | $\text{ND}_{\text{sum}}$ | $\text{NRMSE}_{\text{sum}}$ | $\text{ND}_{\text{sum}}$ | $\text{NRMSE}_{\text{sum}}$ | $\text{ND}_{\text{sum}}$ |
| Transformer-MAF [5] | $0.634 \pm 0.034$ | $\mathbf{0.323 \pm 0.031}$ | $0.039 \pm 0.00$ | $0.030 \pm 0.00$ | $0.363 \pm 0.00$ | $0.301 \pm 0.02$ |
| TimeGrad [6] | $0.715 \pm 0.046$ | $0.399 \pm 0.023$ | $0.039 \pm 0.00$ | $0.026 \pm 0.00$ | $0.073 \pm 0.00$ | $0.055 \pm 0.00$ |
| **ProTran (Ours)** | $\mathbf{0.579 \pm 0.050}$ | $\mathbf{0.317 \pm 0.027}$ | $\mathbf{0.030 \pm 0.00}$ | $\mathbf{0.022 \pm 0.00}$ | $\mathbf{0.046 \pm 0.01}$ | $\mathbf{0.031 \pm 0.00}$ |

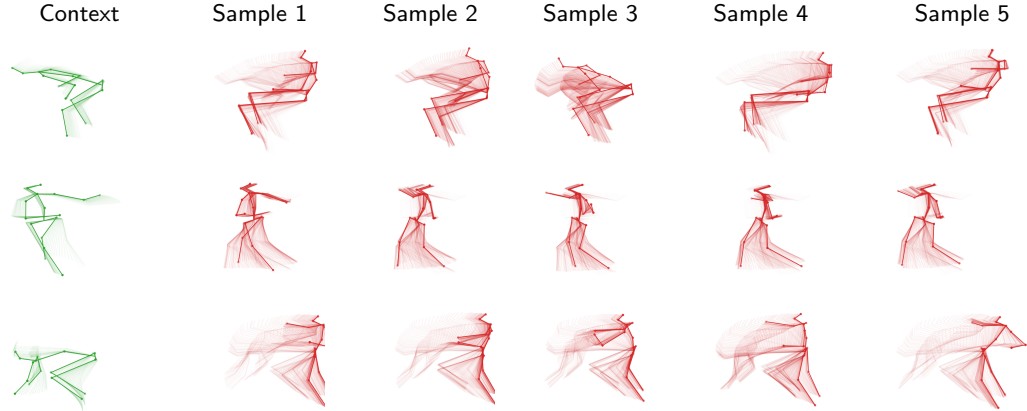

Figure 1: Conditioning pose sequences in green and corresponding predictions in red by ProTran. Solid colors indicate later time-steps and faded ones are older.

## B.2 Human Motion Prediction

Figure 1 show that given the same contexts consisting of fixed conditioning pose sequences, our model is capable of generating diverse yet sensible pose sequences. The variations in predictions stem from the stochasticity induced by our latent variables at different timesteps.

## C Model Architectures

As described in Section 5, our models are based on transformer architectures [9] with extensive use of attention modules. For all experiments, we use a single linear layer to map inputs into fixed-size representations in $\mathbb{R}^{128}$ or $\mathbb{R}^{256}$) (see Equation (5)). We use multihead attention with 8 heads in Equation (6), (7), and (10) to model temporal interactions between latent variables, dependencies on conditional inputs, and interactions of all inputs in posterior distributions. The MLPs in Equation (8) and (11) as well as the final MLP that maps latent variables to outputs consist of 2 layers each with ReLU or Tanh activations. We use fixed positional embeddings as in [9] (see Equation (5) and (9)). The LayerNorms in Equation (5), (6), (7), (9) all have learnable parameters with $\epsilon = 10^{-5}$.

For time series forecasting, we also employ a learnable embedding layer, the outputs of which are concatenated with the lagged inputs as in [5]. Our objective function (see Equation (3)) has an $L_1$ reconstruction loss in most cases, except for the TRAFFIC dataset, in which case we replace it with binary cross entropy and enforce outputs to be in the $[0, 1]$ domain.

Table 3 and 4 detail the numbers of parameters of our models in time series and human motion experiments, respectively. In all cases, our model sizes are comparable or smaller than other baselines.

Table 3: Number of parameters of Transfomer-MAF [5], TimeGrad [6], and ProTran (our model) used in the time-series forecasting experiments.

| DATASET | SOLAR | ELECTRICITY | TRAFFIC | TAXI | WIKIPEDIA |
|---|---|---|---|---|---|
| Transformer-MAF | 290,181 | 532,734 | 1,150,047 | 1,333,706 | 2,229,500 |
| TimeGrad | 116,959 | 300,216 | 1,010,691 | 1,126,974 | 3,099,501 |
| ProTran | 342,418 | 464,292 | 844,998 | 695,612 | 1,510,496 |

[1]`https://archive.ics.uci.edu/ml/datasets/ElectricityLoadDiagrams20112014`
[2]`https://archive.ics.uci.edu/ml/datasets/PEMS-SF`
[3]`https://www1.nyc.gov/site/tlc/about/tlc-trip-record-data.page`
[4]`https://github.com/mbohlkeschneider/gluon-ts/tree/mv_release/datasets`

Table 4: Number of parameters of DLow[10], its conditional VAE model, and ProTran (our model) used in the human-motion prediction experiments.

| DATASET | HUMAN3.6M | HUMANEVA-I |
|---------|-----------|------------|
| CVAE    | 725,292   | 717,174    |
| DLow    | 2,763,820 | 2,753,398  |
| ProTran | 1,166,704 | 1,163,626  |

# D   Hyperparameters & Training Details

For all experiments, we use a batch size of 64 and train for a maximum of 300 epochs with learning rate $3 \times 10^{-4}$. We use Adam optimizer with default parameters from PyTorch and optionally use exponential moving average with rate 0.99. The constant $\beta$ described in Section 2 is fixed for most experiments at 1.0, except for the HumanEva-I in which we use $\beta = 10^{-2}$. We train our models on NVIDIA 2080Ti GPUs. Table 5 shows that our model is comparable to other baselines in terms of running time and testing time.

Table 5: Average training time per epoch and average testing time (in second) of Transfomer-MAF [5], TimeGrad [6], and ProTran (our model) on the TRAFFIC dataset.

| Method | TRAINING | TESTING |
|--------|----------|---------|
| Transformer-MAF [5] | $8.524 \pm 0.001$ | $17.835 \pm 0.002$ |
| TimeGrad [6] | $30.128 \pm 0.002$ | $44.171 \pm 0.003$ |
| **ProTran (Ours)** | $25.832 \pm 0.001$ | $0.168 \pm 0.001$ |