# OpenReview forum: "Probabilistic Transformer For Time Series Analysis"
_NeurIPS.cc/2021/Conference — NeurIPS 2021 Poster_

### Official Review · Reviewer_KKhX · 2021-07-01

**Rating:** 8
**Confidence:** 4

**Summary:**

This paper introduces a deep latent variable model for temporal data, that uses a transformer-based architecture with an attention mechanism over the latent states of a SSM.
This model can then be seen as a probabilistic transformer, and outperforms a wide range of competing models in time series forecasting and human motion prediction experiments.


**Limitations And Societal Impact:**

Yes

**Main Review:**

I found this paper to be very interesting. It presents a novel architecture that can be used for time series modelling, which combines the expressiveness of transformer-based architectures with the ability to model uncertainty of deep SSMs. These are both popular research directions that have had many recent successes in a wide range of applications, so I was happy to see a work trying to get the best of both worlds. The paper is well written and presents an extensive literature review.

Unlike several competing methods, this model relaxes the markovian assumption and provides a natural way to compute the smoothed posterior distribution. Also, the transformer allows to better model long term dependencies in the data, that are hard to capture for RNN-like models

My main concern for the paper is the lack of convincing ablation studies, which I deem a must to really understand what makes such a complex architecture work. The only ablation study is performed on a single dataset, so it is hard to tell if the findings generalize. In particular it would be interesting to understand

1. How important is it to have latent states in the model, as opposed to a fully deterministic architecture? From equations (9) and (20) it is clear that the model could easily learn to ignore all stochasticity. The fact that you pick beta 1e-2 in the Human Eva-I dataset also hints that maybe stochasticity is not that necessary.

2. How important is it to include a latent state at each time step, as opposed to e.g. the models combining transformers and VAEs that you mention in the related work?

3. What is the impact of the number of stochastic layers in different datasets?

4. Why is the laplace distribution a better choice than the more commonly used Gaussian distribution?

Minor comment:
* The ablation study paragraph in line 248 should go in the previous section (5.1)


_____________________________
Reply to author rebuttal:
Thanks for you rebuttal, I have raised my score and will argue for acceptance.

**Time Spent Reviewing:**

4

---

> ### Author Response · Authors · 2021-08-11
> **Responses to Reviewer KKhX**
>
> We would like to thank the reviewer for the positive assessment of the paper. We agree with the reviewer that our ablation studies are still quite limited in scope and will expand the analyses on more datasets to better understand the model components. We also note that evaluating stochastic predictions is challenging and that the existing metrics are not necessarily adequate for such a purpose yet.
>
> The following answers to the questions above are based on the existing results and our experiences.
> * The inclusion of latent variables is important for our models to capture uncertainty, especially for datasets with a lot of possible futures given any conditioning past such as Human3.6M. Deterministic approaches often produce blurry averages of all possibilities.
> * Compared to models that make use of a global latent variable instead of time-dependent ones, our models tend to output more diverse predictions (see Figure 1 in Appendix B). Similar observations have been reported for the task of video prediction [1].
> * The ablation studies do not appear to show significant impacts of the number of stochastic layers, and we have only tested with a few layers due to the linear increase in time complexity. However, we hypothesize that with more efficient transformer techniques, such effects will more visible as we add more stochastic layers, especially for complex domains such as videos. Recent work on deep VAEs [2, 3] has testified to such a direction.
> * Empirically, we find that the Laplace distribution is easier to work, and it probably has something to do with its lighter tail.
>
>
> [1] Denton, E., & Fergus, R. (2018, July). Stochastic video generation with a learned prior. In International Conference on Machine Learning (pp. 1174-1183). PMLR.
> [2] Ho, J., Jain, A., & Abbeel, P. (2020). Denoising diffusion probabilistic models. arXiv preprint arXiv:2006.11239.
> [3] Child, R. (2020). Very deep vaes generalize autoregressive models and can outperform them on images. arXiv preprint arXiv:2011.10650.

---

### Official Review · Reviewer_jdmt · 2021-07-13

**Rating:** 5
**Confidence:** 4

**Summary:**

The submission presents a probabilistic extension to the Transformer by augmenting the intermediate Transformer representations via stochastic latent variables. The attention mechanism allows for accessing the past and the future steps directly in contrast to the indirect hidden state of RNN-based state-space models (SSMs). The building blocks of the Transformer remain untouched. The attention operation is applied to the deterministic representations as in the Transformer. The random variables are then calculated by using these updated deterministic representations. The authors also propose a hierarchical variant of their model to enable stacking several layers as commonly observed in the Transformer architectures.

**Limitations And Societal Impact:**

The authors briefly mention a potential misuse, namely the surveillance scenario, however, they do not discuss how this risk can be mitigated.

**Main Review:**

Neural state-space models (SSM) have been an active domain and stochastic latent variables have been introduced in a vast range of network architectures including but not limited to fully connected, convolutional, rnn, bi-rnn, tcn. Similarly, this paper combines ideas from two different domains, namely the SSMs and the Transformers. The paper does not present a novel method, yet the proposed hybrid architecture seems effective (based on the provided experiments).

It is well-written and easy to follow. It provides a good overview. The design choices are motivated. I enjoyed reading the paper. The formulation is correct. The authors also provide experimental details that are enough to reproduce the results as well as the code.

The authors introduce stochastic latent variables in a straightforward manner. While this simplicity is an advantage, it also impairs the underlying model. For example, the recursive nature of the latent variables takes away the inherent parallelism of the Transformer.

The discussion about the limitations of the RNN-based SSMs is accurate (#33-#36). However, the authors ignore the convolutional SSMs (see [2] and [3]) which alleviate this problem and have similar motivation with the submission. It would be useful to include a discussion involving them.

I am a bit confused about the hierarchical extension. In equations 12-20, the prior model can access the future information through the lower latent variable in the hierarchy, $z_{1:T}^{(l-1)}$. Can the authors clarify this? Similarly, the prior term condition at #155 and in Eq. (15) should be $z_{1:T}^{(l-1)}$ as in Eq. (12)? The references for hierarchical VAEs (#146) should include [2, 3, 4].

#130-#132: The statement should be corrected as [30] is not limited to filtering. Similarly, [1] considers future information in the approximate posterior model via a bi-directional RNN.

The experiment section lacks an in-depth discussion. The authors could explain the motivation for selecting specific baselines. Other than beating several baselines, what does the time-series forecasting (sec 5.1) experiment tell us? The motion prediction experiment provides both quantitative and qualitative analysis. The qualitative results are useful.
In the experiments, did the authors use the prior or the approximate posterior model? Both experiments could include the vanilla Transformer baseline.


References
[1] Goyal et al., Z-Forcing: Training Stochastic Recurrent Networks
[2] Aksan et al., STCN: Stochastic Temporal Convolutional Networks
[3] Lai et al., Stochastic wavenet: A generative latent variable model for sequential data
[4] Gulrajani et al., PixelVAE: A latent variable model for natural images
[5] Martin et al., The Monte Carlo Transformer: a stochastic self-attention model for sequence prediction

---------------------
**Update:**
I thank the authors for their response. I decided to keep my rating. The submission presents a solid formulation and better results in the experiments. The experimental setup has been my main concern. As the main contribution is a Transformer-based SSM, the baselines should involve the established RNN- and CNN-based SSMs as well as the vanilla Transformer. Only one of the two experiments focuses on this setup, but it also does not compare with some of the SOTA SSMs. I suggest the authors introduce the SSMs into the motion experiment baselines as it could be a new and challenging benchmark for the SSMs.

**Time Spent Reviewing:**

4

---

> ### Author Response · Authors · 2021-08-11
> **Responses to Reviewer jdmt**
>
> We would like to thank the reviewer for the constructive comments and relevant references. As mentioned, our proposed models combine existing ideas and are among various latent variable models; however, we emphasize that most previous neural state space models do not focus on time series applications and the community can benefit more from recent advances in additional to classical techniques.
>
> Please see our responses below.
> * While the recursive nature of the latent variables indeed hinders parallelism, we note that such a choice is not only for simplicity but also for performance. When we remove the recursiveness in each layer and rely on interlayer connections to model temporal dependencies, temporal consistency is often not observed in practice. Consequently, we need to sample the latent variables sequentially and conditionally on each other.
> * We appreciate that the reviewer brings convolutional SSMs to our attention and have included the references in our revision. While these approaches also achieve impressive results and share similar motivations, our paper employs transformer architectures, which are quite different from WaveNet, and focuses on time series applications rather than text or speech.
> * Please accept our apologies for the typo in Equation (15); $\\mathbf{z}_\{1:T\}^\{(\ell - 1)\}$ should be in place of $\\mathbf{z}_\{1:T\}^\{(\ell)\}$. Here, we allow the top layers to access the inferred latent variables from the lower layers, including those corresponding to future time steps, to increase model expressiveness.
> * We agree with the reviewer that [1] is not limited to filtering and have corrected the reference on line 131.
> * As suggested by the reviewer, we have added more in-depth discussions on the experiment section and included more qualitative results to further demonstrate that our models are capable of generative accurate, yet diverse predictions thanks to the combination of SSMs and Transformer.
> * At test time, we use the inference model to infer the latent variables $\\mathbf{z}_\{1:C\}$ corresponding to the conditioning sequence $\\mathbf{x}_\{1:C\}$ which are then taken as inputs by the generative model to predict future states $\\mathbf{z}_\{C+1:T\}$ and future observations $\\mathbf{x}_\{C+1:T\}$.
> * We want to point out that we do include time series forecasting results for vanilla Transformer in Table 1 (see Transformer-MAF).
>
> [1] Goyal, A., Sordoni, A., Côté, M. A., Ke, N. R., & Bengio, Y. (2017). Z-forcing: Training stochastic recurrent networks. arXiv preprint arXiv:1711.05411.

---

### Official Review · Reviewer_J2rG · 2021-07-18

**Rating:** 9
**Confidence:** 5

**Summary:**

The paper proposes a probabilistic forecasting model combining transformers with state space models in a non-trivial fashion. It uses variational inference for inference of the multiple, hierarchically stacked state variables (instead of other related works that assume linear state transitions and can then use kalman filtering). The empirical evaluations show the practical relevance of this approach.

**Limitations And Societal Impact:**

yes

**Main Review:**

This is an impressive paper overall on which I can only commend the authors. The proposed model brings together a number of interesting techniques in a non-trivial way: transformers, state-space models (non-linear state transitions), VAEs. The paper discusses relevant work authoritatively, clearly and crisply. The description of the model and inference is clear. However, I am most impressed with the empirical evaluations. The authors picked relevant and multiple methods for comparisons on standard data sets across a spectrum of recent methods. I very much wish that more papers would follow this role-model approach.

I really don't have major comments apart from this. Surely, the authors could have worked more on the well-known short-comings in terms of computational efficiency of transformers (they comment on this in l. 291 thoughtfully), but given the breadth and depth of the contributions of this paper, it's really not necessary. It would have been interesting to see whether the methods in Table 1 are re-implementations or which implementations were chosen and I hope that the authors will release the code in open source.

l. 53. From the notation it is unclear whether x_i \in \mathbb{R}^N. It should be (x_1, \ldots, x_C) \in \mathbb{R}^{C \times N}

**Time Spent Reviewing:**

6h

---

> ### Author Response · Authors · 2021-08-11
> **Respones to Reviewer J2rG**
>
> We thank the reviewer for the careful consideration and encouraging feedback. Please see our responses below.
> * The reviewer is correct that the line 53 should read $\mathbf{x}_{1:C} = (\mathbf{x}_1, \mathbf{x}_2, \dots, \mathbf{x}_C)$ where $\mathbf{x}_t \in \mathbb{R}^N$ for $t \in [1, C]$. Please accept our apologies for the typo.
> * We also agree that improving the time complexity of our models is the next important step. As mentioned in Section 6, we are actively investigating sparse transformer techniques for applications with very long-range dependencies in the inputs. That said, the time complexity do not seem to be a bottleneck for typical sequential prediction tasks such as time series forecasting.

---

### Official Review · Reviewer_m6pu · 2021-07-19

**Rating:** 6
**Confidence:** 4

**Summary:**

## Summary
The authors propose a probabilistic generative non-autoregressive transformer based state space model (SSMs) and corresponding inference procedures for them. Starting with a detailed description of the related probabilities within their models. Followed by a description of the architecture split up into Generative Model and Inference Model also serving a Multi-Layered Extension. In related work several alternative forecasting method types are presented. They compare their methods with 11 of these methods.

**Limitations And Societal Impact:**

Yes

**Main Review:**

## Pros
- detailed preliminaries
- very detailed mathematical declaration of how the different probability distributions form from the given ones
- showing memory and runtime complexity of different models

## Cons
- Incomplete  method formalization (parameter initialization, information/processing flow)
- Partly novel, but limited significance
- More general discussion and implications would be desirable


## Detailed Issues

- Hard to follow preliminaries (fundamentals) section since the authors do neither reference back to previous work nor give a high-level description of the context of their work. Figure 1 is not really helpful since it does not show the parameters of the proposed architecture.
- The concept of distribution forecast is not properly introduced
- Some well annotated block diagrams to explain the architecture and the relation between different
variables would be desirable
- misses a proper explanation of how w’s and z’s are initialized
- Not really using a transformer but multihead attention and positional embedding in some customized way and a encoder decoder structure
- Maybe the  related work section should be move to the front of the paper
- table containing models and parameters used for experiments (as a central and clear overview) would be nice
- unclear when inference model and when generative model is used (one for training one for inference?)

## Minor Flaws
Figure 1: missing connections to the x states
53: (x1, x2, …, xC) \in R^N is wrong because x1 \in R^N
58: could use some references?
77: vague statement with over 1000 page reference.
91 - 94: c is not really a large constant in (attention is all you need
(1/10000) and i should be 2i)
104: not really the transformer architecture (as stated before) rather
using pos-embedding and multi-head-attention with a custom MLP as encoder
Equation 8: z_t depends directly on z_t?
115: somewhat contradicts with conclusion (line 294)
142-143: grammar
167: keeps statt keeps und filter state filer (typos)
177: transformers listed under Attentive Recurrent Networks
Table 2: some caption would be nice
217: different citation style
251-252: what is the actual statement
216 (CRPS equation): what are x and z here and what means {x<=z}
298-301: What is the actual statement/message here?

**Time Spent Reviewing:**

4

---

> ### Author Response · Authors · 2021-08-11
> **Responses to Reviewer m6pu**
>
> We thank the reviewer for the thorough review and thoughtful suggestions. We have revised our paper based on the feedback and improved its readability, although it doesn't seem possible to upload the revision here.
>
> It is important to note that while the model components such as state space models or transformers are not new, combining the ideas into an expressive probabilistic model with tractable inference mechanism is nontrivial and novel to the best of our knowledge. We are mostly interested in forecasting applications in this paper and have shown significant improvements over multiple competitive baselines. Our approach also has potentials when it comes to complex domains such as videos, where probabilistic modelling and powerful architectures are both important.
>
> Our responses to the detailed issues are below.
> * As suggested by the reviewer, we have expanded the preliminary section to include more intuition and contexts on state space models. While we do include references to relevant work in this part, our intention is to provide a succinct and self-contained summary of model assumptions and model components inspired by previous work.
> * While we agree with the reviewer that Figure 1 probably deserves more explanation, we think it helps describe our models using standard graphical notations and draw contrast with linear dynamical systems (LDSs). Similar representations are also quite common in previous work that employs latent variables [1, 2].
> * We have added a sentence describing the differences between distribution forecasts and point forecasts as suggested by the reviewer.
> * We would add a block diagram to help visualize the model architecture upon acceptance, although we think Equations (6)-(20) have also described the architecture precisely in detail.
> * We use Xavier initialization to initialize model weights, including the learning representation for $\mathbf{w}$, and have added the details in Appendix D.
> * The numbers of parameters for all experiments have been previously included in Table 3 and Table 4 in Appendix C.
> * As in other VAE-based models, our approach consists of a generative model and an inference model, which are trained simultaneously and run during training and testing. Inference also happens at test time as we infer the latent variables $\\mathbf{z}_\{1:C\}$ corresponding to the conditioning sequence $\\mathbf{x}_\{1:C\}$.
>
> Regarding the minor flaws, we really appreciate the reviewer’s feedback and has incorporated some changes. In particular, the reviewer is correct that line 53 should read $\mathbf{x}_{1:C} = (\mathbf{x}_1, \mathbf{x}_2, \dots, \mathbf{x}_C)$ where $\mathbf{x}_t \in \mathbb{R}^N$ for $t \in [1, C]$ and that the constant $c$ in line 53 is meant to be small. As to continuous ranked probability score (CRPS) defined in Section 5.1, $x$ denotes an observation and $z$ a variable to be integrated over.
>
> [1] Kingma, D. P., & Welling, M. (2013). Auto-encoding variational bayes. arXiv preprint arXiv:1312.6114.
> [2] Murphy, K. P. (2012). Machine learning: a probabilistic perspective. MIT press.

---

### Decision · Program_Chairs · 2021-09-27

**Decision:**

Accept (Poster)

**Comment:**

Two reviewers advocate strongly for acceptance, one reviewer has been convinced to favor acceptance, and one reviewer recommends rejection. I agree with these first two reviewers about the novelty of the proposed method, and I shared their enthusiasm. I share the concern of reviewer jdmt about the omission of a comparison to convolutional SSMs. Because the authors’ empirical results are already fairly extensive and impressive, I do not favor rejection on these grounds, but I do encourage the authors to include a comparison to convolutional SSMs in the camera-ready version of their manuscript.